# Automated Adenoid Hypertrophy Assessment with Lateral Cephalometry in Children Based on Artificial Intelligence

**DOI:** 10.3390/diagnostics11081386

**Published:** 2021-07-31

**Authors:** Tingting Zhao, Jiawei Zhou, Jiarong Yan, Lingyun Cao, Yi Cao, Fang Hua, Hong He

**Affiliations:** 1Department of Orthodontics, Hubei-MOST KLOS & KLOBM, School & Hospital of Stomatology, Wuhan University, Wuhan 430079, China; zhaott1991@whu.edu.cn (T.Z.); yanjiarong@whu.edu.cn (J.Y.); caolingyun@whu.edu.cn (L.C.); 2School of Physics and Technology, Wuhan University, Wuhan 430079, China; zhoujw@whu.edu.cn; 3School of Electronic Information, Wuhan University, Wuhan 430079, China; caoyi@whu.edu.cn; 4Center for Evidence-Based Stomatology, Hubei-MOST KLOS & KLOBM, School & Hospital of Stomatology, Wuhan University, Wuhan 430079, China; 5Division of Dentistry, School of Medical Sciences, Faculty of Biology, Medicine and Health, The University of Manchester, Manchester Academic Health Science Centre, Manchester M13 9PL, UK

**Keywords:** deep learning, cephalometry, adenoid hypertrophy, nasopharynx, neural networks

## Abstract

Adenoid hypertrophy may lead to pediatric obstructive sleep apnea and mouth breathing. The routine screening of adenoid hypertrophy in dental practice is helpful for preventing relevant craniofacial and systemic consequences. The purpose of this study was to develop an automated assessment tool for adenoid hypertrophy based on artificial intelligence. A clinical dataset containing 581 lateral cephalograms was used to train the convolutional neural network (CNN). According to Fujioka’s method for adenoid hypertrophy assessment, the regions of interest were defined with four keypoint landmarks. The adenoid ratio based on the four landmarks was used for adenoid hypertrophy assessment. Another dataset consisting of 160 patients’ lateral cephalograms were used for evaluating the performance of the network. Diagnostic performance was evaluated with statistical analysis. The developed system exhibited high sensitivity (0.906, 95% confidence interval [CI]: 0.750–0.980), specificity (0.938, 95% CI: 0.881–0.973) and accuracy (0.919, 95% CI: 0.877–0.961) for adenoid hypertrophy assessment. The area under the receiver operating characteristic curve was 0.987 (95% CI: 0.974–1.000). These results indicated the proposed assessment system is able to assess AH accurately. The CNN-incorporated system showed high accuracy and stability in the detection of adenoid hypertrophy from children’ lateral cephalograms, implying the feasibility of automated adenoid hypertrophy screening utilizing a deep neural network model.

## 1. Introduction

Located in the posterior and anterior wall of the nasopharynx, the adenoids are parts of the pharyngeal lymphoid ring. The adenoids, or pharyngeal tonsils, increase in size during childhood to twice of their final adult size with a particular pattern of growth. Under the physiological condition, adenoids often get smaller at the age of 6 and disappear at 10 years old. However, frequent upper airway infections can lead to pathological hypertrophy of the adenoids. The prevalence of adenoid hypertrophy (AH) in children and adolescents ranges from 42 to 70% [1]. AH is one of the most prevalent causes of upper airway obstruction and obstructive sleep apnea (OSA) in children [2].

Mouth breathing resulted from upper airway obstruction may lead to abnormal dentofacial development. Many previous studies have focused on the association between mouth breathing and dentofacial development, according to which mouth breathing could lead to narrow upper arch, longer facial height, steeper mandibular plane angle, and a more retrognathic mandible [3,4]. In addition, failure to thrive, neurobehavioral problems, and depressive symptoms are also believed to be associated with pediatric OSA [5,6,7,8].

Children with AH usually present in orthodontics department with malocclusion, thus the routine screening of AH in dental practice is helpful for preventing relevant craniofacial and systemic consequences [9]. Nasal endoscopy stands as the current gold standard of diagnosing AH [10]. However, nasal endoscopy is painful, and some young children cannot cooperate adequately. Plenty of studies have been performed to identify other reliable diagnostic tools for the detection of hypertrophic adenoid. In orthodontic practice, the lateral cephalogram is a simple, economic, and routine examination. Many studies have proven that lateral cephalograms had high reliability in detecting AH [11,12]. Recently, a systematic review suggested that despite a relatively high false-positive rate, the lateral cephalogram has great diagnostic accuracy (area under the receiver operating characteristic curve = 0.86) for the diagnosis of AH [13].

One of the most notable AH assessment method based on cephalograms is Fujioka’s adenoid–nasopharyngeal (AN) ratio [14]. In Fujioka’s [14] assessment method, four relevant landmarks are manually marked on the cephalograms to measure the AN ratio, which is similar to the process of cephalometric analysis. However, the entire assessment process, including landmark identification, is highly time-consuming and involves repetitive work. Besides, the accuracy of landmark identification depends largely on the examiner’s clinical experience. Inaccurate identification of cephalometric landmarks may lead to incorrect assessment results. Therefore, it is necessary to develop an accurate and efficient algorithm to automatically classify AH in lateral cephalograms.

Artificial intelligence (AI) refers to intelligence demonstrated by machines that can imitate human knowledge and behavior. Deep learning is a subtype of machine learning technique using multi-layer mathematical operations for automated learning and inferring complex data, such as imagery [15]. Deep learning structures, such as convolutional neural networks (CNNs), have been widely used for automatic image classification [16]. In dentistry, images play an important role in screening, diagnosis, and treatment planning. Moreover, the application of deep learning algorithms in cephalometric analysis and the diagnosis of skeletal classification has shown good performance [17,18,19,20]. However, research on the use of deep-learning-based methods in radiographic AH assessment is still limited [21].

Therefore, the purpose of this study was to propose a deep learning method for automated AH assessment based on lateral cephalograms.

## 2. Materials and Methods

This study was approved by the Ethics Committee of the School and Hospital of Stomatology, Wuhan University (No. 2020-B55).

### 2.1. Samples and Identification of Landmarks

The pre-treatment digital lateral cephalograms of all outpatients (6 y to 12 y, *n* = 937) attending the Department of Orthodontics, Hospital of Stomatology, Wuhan University in April–August, 2019 were collected. As determined a priori, 36 images with poor quality, including those with unclear occipital slope, were excluded, resulting in a sample of 901 cephalograms (normal: 651, moderate hypertrophy: 197, severe hypertrophy: 53). The method used for AH assessment was based on Fujioka’s A/N ratio [14]. As shown in Figure 1a, line segment L is drawn along the straight part of the anterior margin of the basiocciput; A’ is the point of maximal convexity along the inferior margin of the adenoid; PNS is the posterior superior edge of the hard palate; line segment A indicates the size of the adenoid, and line segment N indicates the size of the nasopharyngeal space. A child can be suspected of AH if the A/N ratio is greater than 60%.

Among the 901 lateral cephalograms, 581 were randomly selected for training, and 160 were randomly selected for validation, while the remaining 160 were used for testing. As shown in Figure 1b, four landmarks (Ba, Ar, A’, PNS) were accurately identified in the training set (*n* = 581) by two well-trained orthodontists (T.Z. and H.H.) together simultaneously and in consent. Ba is the most inferior-posterior point on the margin of the foramen magnum; Ar is the intersection of the inferior cranial base surface and the averaged posterior surfaces of the mandibular condyles.

Given that the original dataset size was relatively small, we augmented the training dataset to improve the performance and generalization ability of the neural network [22]. The original images were rotated from −20 to 20 degrees around the image center. In addition, these images were shifted by 10 pixels in the up, down, left, and right directions, and 20 pixels in the diagonal directions. The rotation and translation processes were carried out in a manner such that the ROI would be always within the image to avoid information loss. After this step, the size of training dataset grew from 581 images to 9877 images.

### 2.2. Model Architecture and Losses

Figure 2 and Table 1 demonstrate the overall architecture of our model, named HeadNet. It consisted of convolutional layers, attention residual modules [23,24], hourglass modules [25], and an integral regression layer [26]. The hourglass module with top-down and bottom-up design built with regular residual module (Appendix A) had the advantage in integrating multiscale information for further detection. The attention residual module (Appendix A) evolved from a regular residual module that was composed of a serialized placed channel attention part (Appendix A) and a spatial attention part (Appendix A) before output, as this kind of combination has been reported to achieve better results [23].

For efficiency considerations, all images (format: JPEG) were resized into the resolutions of 256 × 256 from 2300 × 2300 without unduly compromising their accuracy. An integral regression layer was applied over generated feature heatmaps by hourglass module to convert them into continuous coordinates [26]. The backpropagation was performed with different losses. The basic loss item was obtained through the comparison between detection and ground truth with L1 Loss, as it performed better than L2 Loss [26].

By deploying prior knowledge for the neural network, the network model could achieve higher performance [27]. Rotation case (Appendix A) would affect the vertical intersection between A’ and Ar-Ba. Translation case (ideal case: Appendix A), which usually comes with rotation (Appendix A), would affect the A and N. The distance from Ar(dt) to ground truth line (formed by ${\mathrm{Ar}}_{\mathrm{gt}}{\;\mathrm{and}\;\mathrm{Ba}}_{\mathrm{gt}}$) is marked as $D_{a}$; the distance from Ba(dt) to ground truth line is marked as $D_{b}$. Intermediate supervision was adopted since it would improve the accuracy of classification [22,28].

To evaluate the effect of our proposed losses, ablation experiments were performed: HeadNet was trained with rotation, translation loss, and attention residual module.

### 2.3. Training Details

We trained the HeadNet with batch size as 10 using the SGD optimizer (momentum was 0.9, and the weight decay was $2\times 10^{-5}$), and all parameters of convolutional layers were initialized randomly. The training process started with warm-up (initial learning rate is 0.001) and an annealing strategy in which the learning rate was updated every 5 epochs.

### 2.4. Statistical Analysis and Evaluation

The absolute distance between the ground truth and the predicted point, and the average precision (AP), as well as the average recall (AR), were the evaluation metrics for keypoint detection. The AN ratio error as the key indicator was the absolute error between the predicted the AN ratio and the actual value. The AN ratio diagnostic accuracy, sensitivity, specificity, receiver operating characteristic curves (ROC), and the area under the curve (AUC), with 95% CIs, were used to test the system’s performance.

## 3. Results

The system showed high performance in AH assessment. The sensitivity, specificity, and accuracy were 0.906 (95% CI: 0.750–0.980), 0.938 (95% CI: 0.881–0.973), 0.919 (95% CI: 0.877–0.961), respectively. The positive likelihood ratio was 10, and the negative likelihood was 0.067. The ROC is provided in Figure 3, and the AUC (95% CI) was 0.987 (95% CI: 0.974–1.000). These results indicated the accuracy of the proposed assessment system.

The evaluation process for 160 sampled images of this diagnostic system took approximately 11 s with a GTX 1070 graphics card. Figure 4 shows changes in the AN ratio error during 200 epochs of training, while Figure 5 shows absolute distance between ground truth and predicted point (in pixel). As the Figure 5 shows, although the average location error is small, the localization error of A’ was exceedingly great, which might be due to unclear adenoid area in validation images. Figure 6 and Figure 7 show changes in the validation of AP and AR during 200 epochs, respectively. These curves suggested that the HeadNet model could learn quickly and find the keypoints location during the first 50 epochs. However, as the model started to converge, the validation error gradually decreased, while validation accuracy increased slowly.

Table 2 presents the performance details of HeadNet. HeadNet * indicates attention residual module was applied. The rotation (r) loss and translation (t) loss were applied in both HeadNet (r, t) and HeadNet * (r, t). HeadNet * (r, t) could achieve the best performance among all the models with F1-Score = 0.936 and AN ratio error = 0.025. Table 3 shows the absolute localization error over keypoints between these models in test dataset; as the table showed, the HeadNetr* (r, t) performs better than other models. Figure 8 shows the predicted keypoints by HeadNet * (r, t) are located closely to the manually landmarked ones.

## 4. Discussion

In children, AH is the most common etiology of partial or complete upper airway obstruction, which can further lead to mouth breathing. Increasing evidence has indicated that AH is associated with dentofacial anomalies [29,30]. For mouth breathing patients, the physiological stimulus for the maxilla growth and the subsequent lowering of the palatal vault could be suppressed due to the reduction of the continuous airflow through the nasal passage [31]. Children with AH are expected to have narrow dental arches, deep palatal height, increased mandibular angle, retrognathic mandible, and convex profile [29,30]. These certain facial features are also called “adenoid facies”.

Both the upper airway and dentofacial structures can be observed in lateral cephalograms, and lateral cephalometry was therefore considered to be a useful screening tool in the assessment of upper airway structures [32,33]. Children with AH usually present in orthodontic clinics with a chief complaint of malocclusion or dissatisfaction with their profile. Besides, the prevalence of pediatric sleep breathing disorder in the general orthodontic population was more than twice that reported in a healthy pediatric population [34]. As cephalometry is routinely performed in orthodontic practice, orthodontists are strongly recommended to screen their patients for sleep breathing disorders and AH in clinical practice [35]. Children with suspected AH based on lateral cephalograms could be referred by orthodontists to the ENT department for diagnosis and treatment [9].

In the present study, we developed an AI method that can assess children’s AH using their lateral cephalograms. The model was trained with lateral cephalograms of pediatric patients and showed the ability of locating the key points for AN ratio. If the AN ratio is greater than 0.6, a diagnosis of AH will be made. Over the 160 test samples, the average keypoint localization error was 1.651 in pixels, while the average accuracy precision, recall, F1 score, and AN ratio error was 0.919, 0.954, 0.936, and 0.025, respectively. The diagnostic accuracy, sensitivity, and specificity were 0.919, 0.906, and 0.938, respectively. Besides, the AUC is 0.99, which far exceeds 0.9. These results indicated that the model was accurate and stable. To our knowledge, so far there are only two studies that have applied AI techniques to AH diagnosis. One of them proposed the VGG-Lite model for the automated evaluation of AH but eliminated the process of landmark identification [36]; the other [21] explored the use of AI in AH diagnosis based on magnetic resonance imaging (MRI), which is not routinely used in orthodontic practice. In contrast, the present study was based on lateral cephalometry, a routine examination conducted by orthodontists. Besides, our AI model was improved to be more suitable for lateral cephalograms and the calculation method. Attention residual modules that we used in this study could apparently improve the performance of keypoints detection and reduce the final AN ratio error.

The significance of this study is that our work could assist clinicians or dentists in the screening of AH by eliminating the possible human errors and greatly reducing the time consumption. Many experienced orthodontists and radiologists can estimate whether the adenoids are hypertrophic just by interpreting the image for a second without measuring the AN ratio. However, it would be time-consuming and fallible when manually evaluating the adenoids of a large sample. Therefore, this automated assessment tool can be used for relevant clinical/epidemiological studies, as well as health examinations at a community/population level.

However, this study has several limitations. Firstly, in order to simplify the labeling and learning process, we used the line connecting points Ar and Ba to replace the line tangent to occipital slope, which is similar to the standard AN ratio measurement method but may result in slightly different results in some borderline cases. Secondly, despite the advantages of being a routine diagnostic tool in orthodontic practice, cephalograms cannot provide 3-dimentional information for either adenoids or the upper airway. A previous study using CBCT showed that a AN ratio >0.6 correlates to a lower nasopharyngeal airway volume but not to the upper airway in general [37]. Thirdly, similar to other dental studies based on cephalograms, we had to manually mark relevant landmarks on cephalograms to construct the reference test [38]. The maximal convexity or deepest concavity on the contour were difficult to identify, which might be the reason why the localization deviation of A’ was relatively large [39].

## 5. Conclusions

The CNN-incorporated system in this study has high accuracy and stability in the detection of AH. AI can be used in the screening of AH among children in dental practice.

## Figures and Tables

**Figure 1 diagnostics-11-01386-f001:**
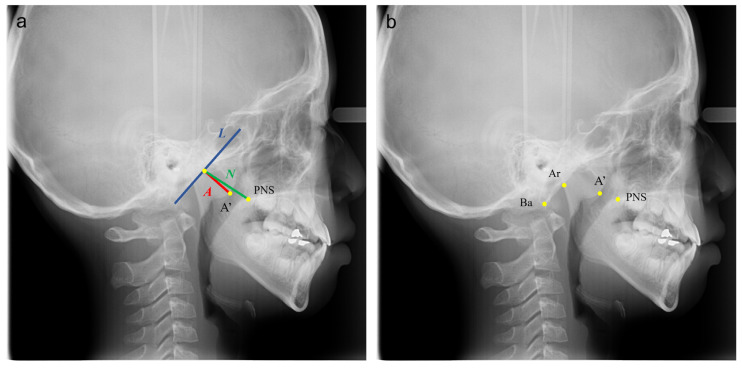
The A/N ratio measurement (**a**); annotated images with four keypoints landmarked (**b**). (A’ is the point of maximal convexity along the inferior margin of adenoid shadow; PNS is the posterior superior edge of the hard palate; Ba is the most inferior-posterior point on the margin of the foramen magnum; Ar is the intersection of the inferior cranial base surface and the averaged posterior surfaces of the mandibular condyles; line segment L is drawn along the straight part of the anterior margin of the basiocciput; line segment A indicates the size of the adenoid; line segment N indicates the size of the nasopharyngeal space).

**Figure 2 diagnostics-11-01386-f002:**
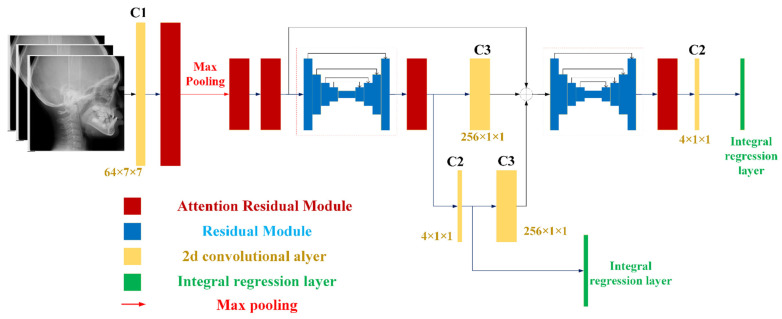
Model Architecture: The yellow rectangle represents the 2-d convolutional layer; the red rectangle represents the attentional residual module; the blue rectangle in the hourglass-style represents normal residual module; the green rectangle represents the integral regression layer, which converts heatmaps into keypoints; each convolutional layer is followed by a ReLU operation.

**Figure 3 diagnostics-11-01386-f003:**
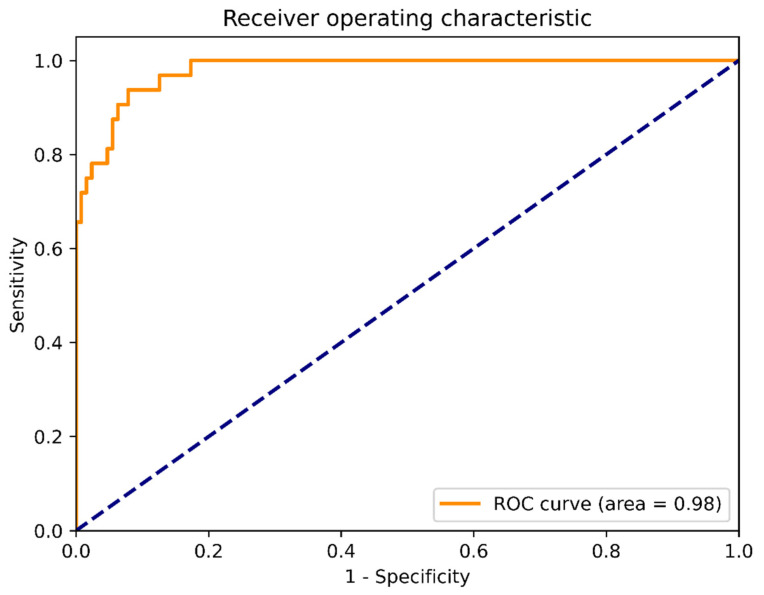
Receiver operating characteristic curve (ROC). The area under the curve (AUC) was far exceeding 0.9, which indicated that the proposed system was able to accurately assess adenoid hypertrophy.

**Figure 4 diagnostics-11-01386-f004:**
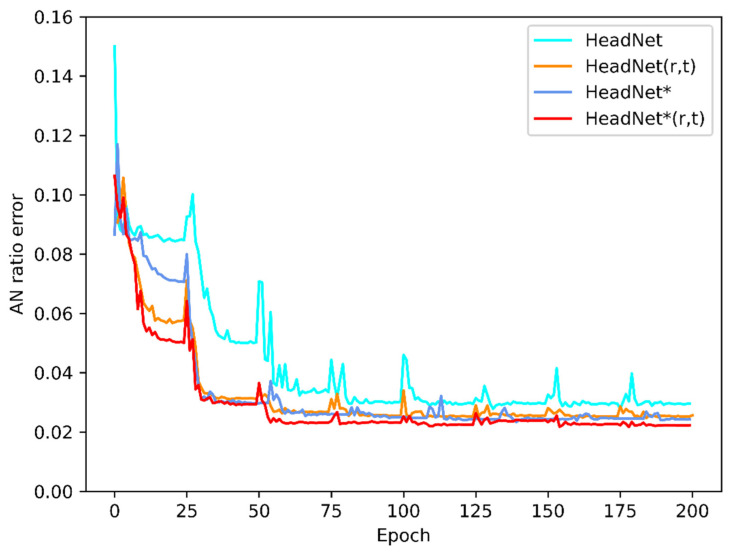
Changes in AN ratio error during model training: AN ratio error decreased as the epochs increased.

**Figure 5 diagnostics-11-01386-f005:**
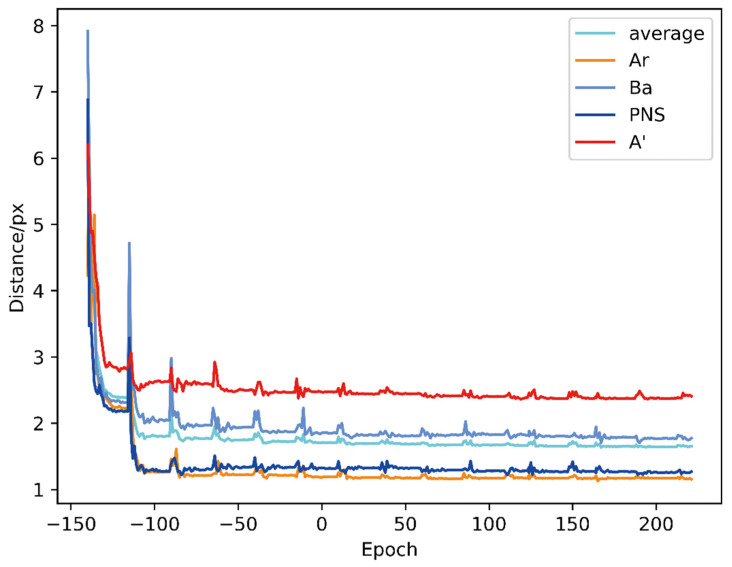
Changes in absolute distance between ground truth and predicted point (in pixels).

**Figure 6 diagnostics-11-01386-f006:**
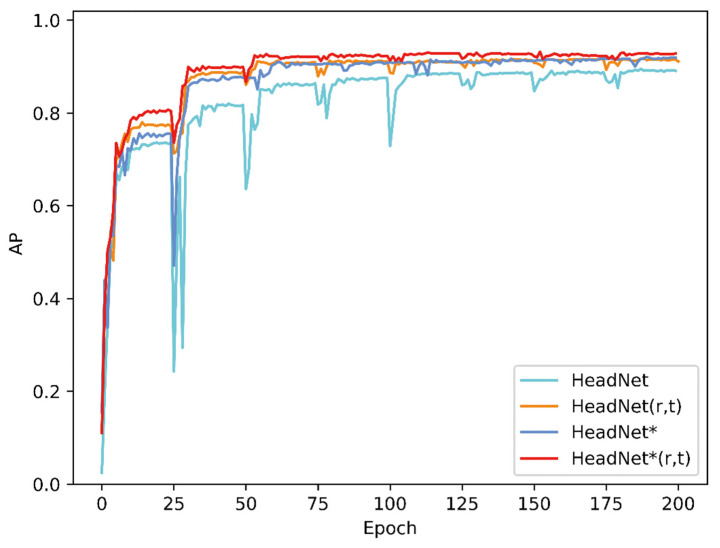
Changes in validation AP of HeadNet.

**Figure 7 diagnostics-11-01386-f007:**
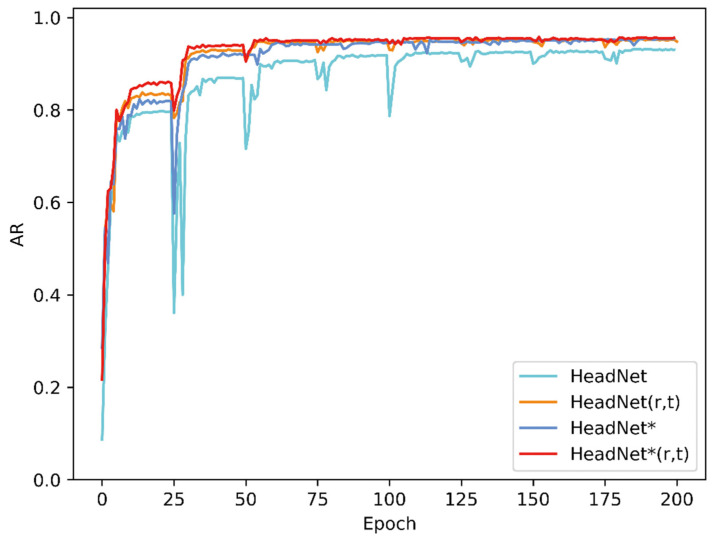
Changes in validation AR of HeadNet.

**Figure 8 diagnostics-11-01386-f008:**
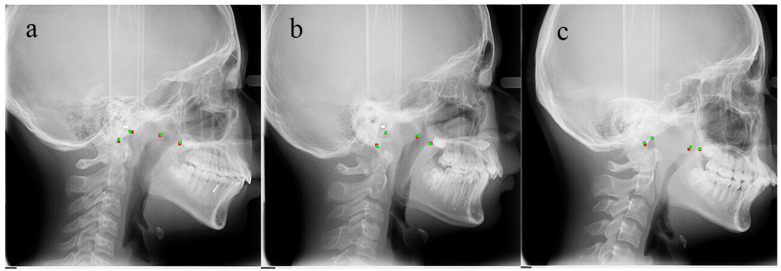
Validation dataset image comparison (green points represent ground truth; red points represent detection). The normal case (**a**); moderate hypertrophic case (**b**); severe hypertrophic case (**c**).

**Table 1 diagnostics-11-01386-t001:** Parameters of convolutional layers in HeadNet.

Name	C1	C2	C3	C4	C5	C6	C7
Output channels	64	4	256	128	128	16	1
Kernel size	7 × 7	1 × 1	1 × 1	1 × 1	3 × 3	1 × 1	7 × 7
Stride	2	1	1	1	1	1	1

C: Convolutional layer; C1, C2, and C3 were used in HeadNet model; C3, C4, and C5 were used in both residual module and attention residual module; C6 and C7 were used in attention residual module.

**Table 2 diagnostics-11-01386-t002:** Performance of HeadNet on test dataset.

Method	AP	F1-Score	A/N Error
HeadNet	0.876	0.896	0.031
HeadNet (r, t)	0.910	0.928	0.027
HeadNet *	0.904	0.923	0.027
HeadNet * (r, t)	0.919	0.936	0.025

r: rotation is applied; t: translation is applied; *: the attention residual module is applied in Headnet.

**Table 3 diagnostics-11-01386-t003:** Absolute localization error (pixel) over keypoints between different models in the test dataset.

Method	Ar	Ba	PNS	A’	Average
HeadNet	1.723	1.961	1.326	2.570	1.895
HeadNet (r, t)	1.285	1.899	1.275	2.575	1.758
HeadNet *	1.276	1.813	1.307	2.416	1.703
HeadNet * (r, t)	1.188	1.744	1.275	2.372	1.651

r: rotation is applied; t: translation is applied; *: the attention residual module is applied in Headnet.

## Data Availability

The data underlying this article will be shared on reasonable request to the corresponding author.

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
