# Peer review of "Automated Adenoid Hypertrophy Assessment with Lateral Cephalometry in Children Based on Artificial Intelligence"

_diagnostics, 2021, doi:10.3390/diagnostics11081386_

Round 1

Reviewer 1 Report

Manuscript No.: diagnostics-1299014 

Title

Automated Adenoid Hypertrophy Assessment with Lateral Cephalometry 

Article Title

The title is adequate

Abstract

Relevant.

Line 25, “is” should read “was”

Introduction

Quit extensive.

There is no discussion of the fact that cephalograms show a 2-dimensional image of the adenoid but the adenoids are 3-dimensional. That should be added here and/or in the discussion.

Line 52; the “A” prior to plenty should eb deleted.

Objectives

The objective is clear

Description of Study Design/ Material and Methods

The study is overall well designed but the following assessment  is not accounted for:

Two experienced orthodontists “accurately identified” the keypoint landmarks. These were used to train the HeadNet. As the keypoint landmarks are the “Gold standard” in this study, it would be interesting to know what the inter- and intra-observer agreement in identifying the landmarks were. Did both of the orthodontists assess the same cephalograms for calculation of interobserver agreement? And did they calculate intra-observer agreement?

If not, I would recommend that this is added to the study protocol and accounted for in the methods and results.

It is a bit confusing that adenoid maximum convexity is marked with an A’ and the distance between A’ and the L-line also is marked with A. The A’ could perhaps be more clearly differentiated from A?

It is very difficult to se the difference in Figure 1.

Results

The results on the system performance are well documented

The interobserver agreement is missing as mentioned above.

Line 209-211 – there is irrelevant text inserted that should be deleted

Discussion

One limitation of the study is addressed in the discussion, concerning the line across the Ar and Ba, saying that it could introduce errors in the classification of AN ratio. Could the authors explain that further?

Some suggestions to be added to the limitations:

  1. The images are 2-dimensional but the adenoid is 3-dimensional.
  2. It should also be clearly mentioned that a patient that doesn't show adenoid hypertrophy might suffer from obstructive sleep apnea (OSA) for other reasons. It is mentioned in the introduction that the adenoid is part of the lymphoid ring (Waldeyers ring). It should be mentioned that both the palatal tonsils and the lingual tonsil can add to obstruction, as well as other anatomical deviations. Hypertrophy of both the palatal and lingual tonsils can sometimes be assessed and suspected on a lateral cephalogram.
  3. Further, the patient might not show OSA despite the 2-dimenional adenoid hypertrophy – if the airway is sufficient in the 3-dimension perspective.
  4. Many experienced orthodontists and radiologists can estimate whether the adenoids are hypertrophic just by interpreting the image for a second without measuring the AN ratio. But of course, they have to remember to look for it.

Perhaps refer to

Feng X, Li G, Qu Z, Liu L, Näsström K, Shi XQ. Comparative analysis of upper airway volume with lateral cephalograms and cone-beam computed tomography. Am J Orthod Dentofacial Orthop. 2015 Feb;147(2):197-204. doi:10.1016/j.ajodo.2014.10.025.PMID: 25636553

The study shows that a AN ratio >0.6 correlates to a lower nasopharyngeal airway volume – but not to the upper airway volume in general when measured with CBCT.

Line 252: “…the measurement mehod they used failed to take keypoints translation into account”. I am not sure what the authors mean by that?

The fact the A’ differed most from “ground truth” – why do you think that was the case? Add a discussion on that?

Conclusions

Corresponds to the objective and result

Figure legends

- could in several cases be more clarifying

References

Adequate – I have suggested an addition of a reference under Discussion.

Author Response

Response to Reviewer 1

Diagnostics-1299014

Article Title

The title is adequate

Response:

Thank you.

Abstract

Line 25, “is” should read “was”

Response:

Thank you very much. Revision done.

Introduction

There is no discussion of the fact that that cephalograms show a 2-dimensional image of the adenoid but the adenoids are 3-dimensional. That should be added here and/or in the discussion.

Line 52 The “A” prior to plenty should be deleted.

Response:

Thank you very much for your suggestions. We have added one paragraph in the Discussion section to show the limitations of the study.

The “A” prior to plenty has been deleted.

Objectives

The objective is clear

Response:

Thank you.

Description of study design/Material and Methods

The study is overall well designed but the following assessment is not accounted for.

Two experienced orthodontists “accurately identified” the keypoint landmarks. These were used to train the HeadNet. As the keypoint landmarks are the “Gold standard” in this study, it would be interesting to know what the inter- and intra-observer agreement in identifying the landmarks were. Did both of the orthodontists assess the same cephalograms for calculation of interobserver agreement? And did they calculate intra-observer agreement?

If not, I would recommend that this is added to the study protocol and accounted for in the methods and results.

Response:

Thank you very much for your suggestions. Actually, in our study, two well-trained orthodontists identified the four keypoint landmarks together simultaneously, not independently. Also, the landmark identification process did not produce any data but only the location of keypoints. Therefore, please understand that we cannot measure inter- or intra-observer agreement. We have revised our manuscript to make this clearer.

It is a bit confusing that adenoid maximum convexity is marked with an A’ and the distance between A’ and the L-line also is marked with A. The A’ could perhaps be more clearly differentiated from A?

It is very difficult to see the difference in Figure 1.

Response:

Thank you very much for your comments. We have revised our figure 1 and the corresponding text to make it easier for readers to understand.

Results

The results on the system performance are well documented

The interobserver agreement is missing as mentioned above

Line 209-211 there is irrelevant text inserted that should be deleted

Response:

Thank you very much. In our study, two well-trained orthodontists identified the four keypoint landmarks together simultaneously, not independently. Also, the landmark identification process did not produce any data but only the location of keypoints. Therefore, please understand that we cannot measure inter- or intra-observer agreement. We have revised our manuscript to make this clearer.

The irrelevant text has been deleted. Many thanks for the reminder.

Discussion

One limitation of the study is addressed in the discussion, concerning the line across the Ar and Ba, saying that it could introduce errors in the classification of AN ratio. Could the authors explain that further?

Response:

Thank you very much for your comments. We have moved this sentence to the Limitations paragraph and revised it to make our point clearer.

Some suggestions to be added to the limitations:

  1. The images are 2-dimensional but the adenoid is 3-dimensional.
  2. It should also be clearly mentioned that a patient that doesn't show adenoid hypertrophy might suffer from obstructive sleep apnea (OSA) for other reasons. It is mentioned in the

introduction that the adenoid is part of the lymphoid ring(Waldeyers ring). It should be mentioned that both the palatal tonsils and the lingual tonsil can add to obstruction, as well as other anatomical deviations. Hypertrophy of both the palatal and lingual tonsils can sometimes be assessed and suspected on a lateral cephalogram.

  1. Further, the patient might not show OSA despite the 2-dimenional adenoid hypertrophy – if the airway is sufficient in the 3-dimension perspective.
  2. Many experienced orthodontists and radiologists can estimate whether the adenoids are hypertrophic just by interpreting the image for a second without measuring the AN ratio. But of course, they have to remember to look for it.

Perhaps refer to

Feng X, Li G, Qu Z, Liu L, Näsström K, Shi XQ. Comparative analysis of upper airway volume with lateral cephalograms and cone-beam computed tomography. Am J Orthod DentofacialOrthop. 2015 Feb;147(2):197-204.doi:10.1016/j.ajodo.2014.10.025.PMID: 25636553

The study shows that a AN ratio >0.6 correlates to a lower nasopharyngeal airway volume – but not to the upper airwayvolume in general when measured with CBCT.

Response:

Thank you very much for your comments and suggestions. We have added one paragraph in the Discussion section to show the limitations of this study.

Please note that the aim of this study was to develop an automated assessment tool for adenoid hypertrophy, not pediatric OSA. We agree that children with AH may or may not have OSA, and have noticed that the previous version of our Introduction failed to point out that AH itself is worth efforts to screen and diagnose. Also, children with suspected adenoid hypertrophy based on lateral cephalograms could be referred by orthodontists to the ENT department for definitive diagnosis and treatment. We have revised our Introduction section accordingly to improve its appropriateness.

We agree with you that the visual assessment of a lateral cephalogram seems enough for an experienced clinician to roughly judge the adenoid condition. However, it would be time-consuming and fallible when manually evaluating the adenoids of a large sample. Therefore, the automated evaluation method has actual significance, it can be used for relevant clinical / epidemiological studies, as well as oral health examinations at a community / population level. We have added several sentences in the Discussion section to make this clear.

Line 252: “…the measurement method they used failed to take keypoints translation into account”. I am not sure what the authors mean by that?

Response:

Thank you very much for your comments. In our study, the rotation loss and translation loss were applied to the training process which would further improve the accuracy of our model. However, the study conducted by Shen in 2020 did not consider translation loss. After consideration, we have deleted this sentence from our article, as this computer science technique related point is very difficult to understand for the readers of this article -- clinicians.

The fact the A’ differed most from “ground truth” – why do you think that was the case? Add a discussion on that?

Response:

Similar to other dental studies based on cephalograms, we had to manually mark relevant landmarks on cephalograms to construct the reference test [PMID: 33631303]. The maximal convexity or deepest concavity on the contour were difficult to identify, which might be the reason why the localization deviation of A’ was relatively large [PMID: 29650356]. To explain this, we have added several sentences and two references in the Discussion section.

Conclusions

Corresponds to the objective and result

Response:

Thank you very much for the recognition of our work.

Figure legends

Could in several cases be more clarifying

Response:

Thank you. Revision done.

References

Adequate-I have suggested an addition of a reference under Discussion.

Response:

Thank you. Revision done.

Reviewer 2 Report

In this manuscript, a tool for adenoid hypertrophy assessment is proposed based on lateral cephalograms. This is a great work, well-written with appropriate structure. The objective is clear and the manuscript of scientific interest. Below are some comments to the authors:

  1. The background is missing from the Abstract. Please include a couple of lines describing in brief the subject related to the paper.
  2. In Abstract the abbreviation SI is used. Please use the first time the whole word and then the abbreviation.
  3. Since the dataset is comprised on pediatric cases I would suggest the authors to include this in the title to be more accurate (e.g. “pediatric”, or “in children”)
  4. In the Discussion only one similar MRI-based study is mentioned. Please include a few more studies utilizing machine learning algorithms to assess adenoid hypertrophy with lateral cephalograms, if any.

Author Response

Response to Reviewer 2

Diagnostics-1299014

Comments and suggestions for authors:

In this manuscript, a tool for adenoid hypertrophy assessment is proposed based on lateral cephalograms. This is a great work, well-written with appropriate structure. The objective is clear and manuscript of scientific interest.

Response:

Thank you very much for the recognition of our work.

 Below are some comments to the authors:

  1. The background is missing from the Abstract. Please include a couple of lines describing in brief the subject related to the paper.

Response:

Thank you very much for your suggestions. We have added two sentences into our Abstract to briefly provide the background.

  1. In Abstract the abbreviation SI is used. Please use the first time the whole word and then the abbreviation.

Response:

Thank you. There is no “SI” in the Abstract. We have checked and spelled out all abbreviations including CI.

  1. Since the dataset is comprised on pediatric cases I would suggest the authors to include this in the title to be more accurate (e.g. “pediatric”, or “in children”)

Response:

Thank you very much. Revision done.

  1. In the Discussion only one similar MRI-based study is mentioned. Please include a few more studies utilizing machine learning algorithms to assess adenoid hypertrophy with lateral cephalograms, if any.

Response:

Thank you very much for your suggestions. Revision done.

Round 2

Reviewer 1 Report

Line 98-100 needs linguistic revision, e.g.,: A’ is the point of the maximum convexity, along the inferior delineation of the adenoid

(don't use “shadow”!)

Line 106 – add how many lateral cephalograms the two experienced orthodontists assessed together

Line 106: The assessment was performed “…..together simultaneously and in consent”.

I suggest that “consent” is added.

Figure 3 – legend: explain what the ROC shows to make it possible for reader to understand the figure without reading the text
